# Fluoxetine Removal from Aqueous Solutions Using a Lignocellulosic Substrate Colonized by the White-Rot Fungus *Pleurotus ostreatus*

**DOI:** 10.3390/ijerph19052672

**Published:** 2022-02-25

**Authors:** Andreia D. M. Silva, Juliana Sousa, Malin Hultberg, Sónia A. Figueiredo, Olga M. Freitas, Cristina Delerue-Matos

**Affiliations:** 1REQUIMTE/LAQV—Associated Laboratory for Green Chemistry (LAQV) of the Network of Chemistry and Technology (REQUIMTE), Instituto Superior de Engenharia do Porto, 4200-072 Porto, Portugal; andreia.silva@graq.isep.ipp.pt (A.D.M.S.); 1191179@isep.ipp.pt (J.S.); saf@isep.ipp.pt (S.A.F.); cmm@isep.ipp.pt (C.D.-M.); 2Department of Biosystems and Technology, Swedish University of Agricultural Sciences, Alnarp, 230 53 Skara, Sweden; malin.hultberg@slu.se

**Keywords:** crude enzyme extracts, laccase activity, mycoremediation, pharmaceuticals, colonized mushroom substrate, tertiary treatment

## Abstract

One of the main challenges in both the design of new wastewater treatment plants and the expansion and improvement of existing ones is the removal of emerging pollutants. Therefore, the search for economic and sustainable treatments is needed to enhance the removal of pharmaceuticals. The potential of a lignocellulosic substrate colonized by *Pleurotus ostreatus*, a waste from mushroom production, to remove fluoxetine from aqueous solutions was studied. Batch assays were performed to remove 600 µg∙L^−1^ fluoxetine from aqueous solutions using the colonized mushroom substrate (CMS) and crude enzyme extracts. The removal efficiencies achieved were, respectively, ≥83.1% and 19.6% in 10 min. Batch assays with sterilized CMS and 1-aminobenzotriazole (to inhibit cytochrome P450 enzymes) showed that the higher removal efficiencies achieved in the CMS assays may be attributed to the synergistic contribution of biosorption onto the CMS and lignin modifying enzymes activity, namely laccase activity. A column assay was performed with the CMS, fed with 750 µg∙L^−1^ fluoxetine aqueous solution. The removal efficiency was 100% during 30 min, decreasing to a final value of 70% after 8 h of operation. The results suggested that CMS can be a promising eco-friendly alternative to remove fluoxetine from aqueous solutions.

## 1. Introduction

Pharmaceuticals (PhCs) are compounds that are highly consumed worldwide and therefore constantly released into aquatic ecosystems as a consequence of human activities and direct discharges from hospitals, pharmaceutical industry, and veterinary facilities, but mainly from urban wastewater treatment plants (WWTPs), which were not specifically designed for PhCs removal [1,2,3].

The release of PhCs into the aquatic environment is not yet subject to regulation. In Europe, under the Water Framework Directive (WFD), the Surface Water Watch List (WL) (2020) was established, which includes PhCs (erythromycin, clarithromycin and azithromycin, amoxicillin, ciprofloxacin, sulfamethoxazole, trimethoprim, venlafaxine and its metabolite O-desmethylvenlafaxine, and a group of ten azole PhCs) that should be carefully monitored by the European Union member states to determine the risk they pose to the aquatic environment. A Groundwater WL was also implemented by the European Union groundwater working group, who defined a Common Implementation Strategy. The Groundwater WL aims to identify the substances for which groundwater quality standards or threshold values should be set. Contrary to the WL for surface water, the monitoring is voluntary. Therefore, it is expected that legal limits will soon be established for the discharge of PhCs into the environment [4] and, simultaneously, a policy to raise public awareness for the correct elimination of excess and/or expired PhCs will be developed.

Even occurring in extremely low concentrations, ranging from µg∙L^−1^ to ng∙L^−1^ or even lower, concerns have been raised over PhCs’ occurrence in the environment because, unlike most pollutants, they are designed to have a specific action on the human body and to act in very low concentrations. Chronic exposure to low concentrations of some PhCs found in the environment can cause adverse effects on human health. Because of these effects, and as PhCs are not commonly monitored or regulated (by national or international legislation), they are categorized as emerging pollutants [5,6,7,8]. There are advanced treatments ready to be used in WWTPs, such as adsorption in activated carbon, membrane filtration, and advanced oxidation processes (e.g., ozonation), but despite their high efficiency, these technologies are often considered expensive and require technical/operational skills and high energy consumption [9]. Aware of this problem, the scientific community has been working to find efficient, economic, and sustainable affination treatment processes for PhCs’ removal [10].

One alternative that has gained increasing interest in recent years is mycoremediation, which is a form of bioremediation in which a fungi-based technology is used. Fungi have been proven to be a very cheap, effective, and environmentally sound way of removing a wide array of PhCs from wastewaters [11]. Despite all the potentialities of mycoremediation, before its full-scale application, limitations such as the need for an additional carbon source, immobilization of fungal biomass, competition with autochthonous microorganisms, and high hydraulic retention times need to be overcome [11].

One of the most promising fungal strains in PhCs removal is the edible mushroom *Pleurotus ostreatus*. Previous studies [10,12,13,14,15] have shown that *Pleurotus ostreatus* can eliminate a wide range of PhCs classes from aqueous solutions. *Pleurotus ostreatus* is a “white-rot fungi” (WRF) basidiomycete. Its ability to degrade lignin and cellulose, the two main building blocks of plant fibre, is due to the secretion of extracellular lignin-modifying enzymes (LMEs) and acids. Extracellular LMEs include versatile peroxidases (VP, Enzyme Commission (EC) 1.11.1.16), manganese peroxidases (MnP, EC 1.11.1.13), and laccases (phenol oxidases, Lac, EC 1.10.3.2). Unlike other WRF, *Pleurotus ostreatus* does not produce lignin peroxidases (LiP, EC 1.11.1.14). The main difference between peroxidases and laccases is the electron acceptor, where hydrogen peroxidase and oxygen are the respective electron acceptors [4,16,17,18]. The composition of the growth medium and culture conditions influence the secretion of a specific enzyme [19,20,21]. Ligninolytic enzymes are usually produced during secondary metabolism. The synthesis and secretion of these enzymes are often induced by the levels of nutrients (carbon and nitrogen); in addition, the production of MnP is optimal in high oxygen concentration but is repressed in submerged liquid cultures, while Lac is favoured [22]. Although less studied, intracellular enzymes may play an important role in the degradation of PhCs. Intracellular enzymes or mycelium-associated enzymes (i.e., cytochrome P450 system (CYP450)) are a group of (mainly) monooxygenases that can degrade PhCs by catalyzing several kinds of reactions, such as heteroatom oxygenation, dehalogenation, dealkylation, epoxidation of C=C, reduction, and hydroxylation [4,12,13,23,24,25]. The WRF *Pleurotus ostreatus* has a CYP450 system that consists of 153 genes [12].

The *Pleurotus* genus is cultivated on a variety of lignocellulosic substrates and various other substrates [26]. After the cultivation, the growing substrate is regarded as waste, known as spent mushroom substrate (SMS). The industrial cultivation of *Pleurotus ostreatus* and other edible mushrooms produces a significant amount of SMS. In Europe, more than 3.5 million tons of SMS are produced annually [27]. The production of an average of 1 kg of fresh mushrooms results in 5 kg of SMS [28,29], being considered an agricultural waste stream. From a circular economy perspective, this agricultural waste is no longer considered a debit entry. Indeed, SMS is currently considered a valuable low-cost and readily available resource for different purposes, such as bioremediation of organic pollutants from water matrices, as it harbours high levels of residual enzymatic activity [30,31,32].

This study aimed to assess the potential of a lignocellulosic substrate colonized by the WRF *Pleurotus ostreatus* to remove fluoxetine, an antidepressant used worldwide, from aqueous solutions and to evaluate the laccase activity of the colonized mushroom substrate (CMS). Even though fluoxetine is consumed in large amounts [33,34], there are few studies about its degradation using fungi. Batch assays were performed under non-sterile conditions to assess the mechanisms involved in the fluoxetine removal, namely the contribution of the biosorption onto the CMS and of intracellular enzymes activity to fluoxetine removal. Fluoxetine removal using crude enzyme extracts (CEE) (extracellular content without CMS) was also studied. The column assay was performed using the CMS to simulate a tertiary treatment process in a WWTP.

## 2. Materials and Methods

### 2.1. Materials

#### 2.1.1. Fungal Strain and Cultivation Conditions

The WRF used in this work was *Pleurotus ostreatus* M2191 strain. The spawn produced on rye kernels was obtained from Mycelia BVBA (Deinze, Belgium). For cultivation, a mushroom substrate, based on alder sawdust (73% of dry weight (dw)), wheat bran (25% of dw), and calcium sulfate (2% of dw), and with a moisture content of 65%, was used. Before inoculation of the spawn, the substrate was packed into gas-permeable bags suitable for mushroom production (Sac O_2_, Nevele, Belgium) and pasteurized at 65 °C for six hours. The spawn was added in a concentration of 10% (dw∙dw^−1^) after the substrate had cooled down. The inoculated substrate bags were then incubated at 22 °C until they were densely colonized by mycelium (fourteen days). The CMS was used directly after fifteen days, being stored cold (10 °C). Before the experiments, the bags were left at room temperature for one day.

#### 2.1.2. Reagents

The pharmaceutical fluoxetine hydrochloride ((RS)-*N*-methyl-3-phenyl-3-[4-(trifluoromethyl)phenoxy]propan-1-amine) (purity > 98%) was obtained from Sigma-Aldrich (Taufkirchen, Germany). The molecular structure of this compound is shown in Figure 1 and its physicochemical properties are listed in Appendix A [35] (Appendix A). The species distribution diagram of fluoxetine as a function of pH is shown in Appendix A [36,37] (Appendix A). Stock standard solution of fluoxetine hydrochloride (1300 mg∙L^−1^) was prepared in a basis weight in methanol and stored at −20 °C in a dark glass vial.

All the working solutions were prepared with HPLC-grade water (resistivity of 18.2 MΩ∙cm^−1^), obtained from a Simplicity 185 system (Millipore, Molsheim, France).

The information concerning the other reagents used is provided in Appendix A.

### 2.2. Methods

#### 2.2.1. Alder Sawdust Characterization

The pH at the point of zero charge (pH_PZC_) and Fourier transmittance infrared (FT-IR) spectrum (Thermo Scientific, Nicolet 6700 FT-IR, MCT/A detector, Waltham, MA, USA) were used to characterize alder sawdust, as described by Silva et al. [38] and Mall et al. [39], respectively. Samples of alder sawdust were ground and dried in an oven (Selecta P, 2000208, Barcelona, Spain) at 70 °C for 24 h and cooled in a silica gel desiccator.

#### 2.2.2. Batch Assays

Batch assays were performed to study the removal of fluoxetine by the CMS and CEE from non-sterile aqueous solutions. The contributions of biosorption and intracellular enzymes (i.e., CYP450) to the fluoxetine removal were also assessed. The assays were performed at room temperature, for 30 min, continuously stirred at 100 rpm (VELP Scientifica, Multistirrer Magnetic Stirrer F203A0178, Usmate Velate, Italy), and at pH 7 (in Britton–Robinson buffer solution), considering the typical pH range for domestic wastewater [40] and the expected range after the secondary treatment (between 6.5 and 7.5) [41]. The assays were performed in 250 mL Erlenmeyer’s flasks with a working volume of 150 mL and an initial concentration of 600 µg∙L^−1^ (>30 times higher than the limit of quantification). An initial sample was immediately collected, and further samples were collected at regular time intervals to determine fluoxetine concentration and laccase activity. To stop the enzymatic activity, 40 µL of sodium hydroxide (NaOH) (2 M) was added to each sample. Before HPLC-FLD analysis, samples were centrifuged at 9000 rpm (Thermo Scientific, Heraeus Fresco 21 Microcentrifuge, Dreieich, Germany) for 10 min, and the supernatants were collected for analysis. The temperature and pH (Consort, C861, Turnhout, Belgium) were recorded at the beginning and the end of the assays. Blank assays were run in parallel in the same conditions. All assays were performed in duplicate.

Two CMS assays were performed with 50 g of the CMS (wet weight, moisture content of 69.9 ± 1.1%) with a fifteen-day lag time (on the first day and after fifteen days of CMS storage, respectively).

The biosorption assays were made with fifty grams of the CMS, previously autoclaved (AJC, Uniclave 88, Portugal) at 120 °C for 20 min.

The CYP450 inhibition assays were performed after fifteen days of CMS storage, using 50 g of the CMS previously mixed with a Britton–Robinson buffer solution (pH 7) containing 1-aminobenzotriazole (1-ABT) (1 mM) for 30 min.

The CEE assays were carried out on the first day of storage, using 50 g of the CMS previously suspended in 150 mL of Britton-Robinson buffer solution (pH 7). The mixture was incubated at 5 °C for at least 2 h and periodically shaken, and then centrifuged at 9000× *g* (Heraeus Megafuge 16R Centrifuge, Thermo Fisher Scientific, Waltham, MA, USA) at 5 °C for 8 min. The supernatant was collected and used as CEE.

#### 2.2.3. Column Assay

After three days of storage, a column assay was performed using a glass column (25 mm inner diameter × 150 mm length; Omnifit^®^, Diba Industries, Cambridge, UK), filled with 12.097 ± 0.001 g of CMS (wet weight, moisture content of 69.9 ± 1.1%, bed volume of 4.3 × 10^−5^ L). A 750 µg∙L^−1^ non-sterile fluoxetine (hydrochloride) solution in Britton–Robinson buffer (pH 7) was fed to the fixed-bed column using a peristaltic pump (Gilson^®^, Minipuls 3, Villiers le bel, France), with a downward flow rate of 3.0 ± 0.3 mL∙min^−1^ (hydraulic flowrate of 4.2 × 10^−3^ ± 4.4 × 10^−4^ m^3^∙min^−1^∙m^−2^), at room temperature, operating for 8 h. Samples were taken at regular time intervals to determine fluoxetine concentration and laccase activity in the outlet solution. Like in batch assays, 40 µL of NaOH (2 M) was added to each sample to stop the enzymatic activity. The temperature and pH of inlet and outlet solutions were also recorded.

#### 2.2.4. Determination of CMS Moisture Content

The moisture content was determined (in triplicate) by weighing 50 g of the CMS (Metter Toledo^®^, New Classic MF, MS 205DU, Greifensee, Switzerland) before and after drying in an oven at 105 °C (Selecta P, 2000208, Barcelona, Spain), for 24 h.

#### 2.2.5. Determination of Laccase Activity

The laccase (EC 1.10.3.2, p-diphenol: dioxygen oxidoreductase) activity was determined by spectrophotometry as described by Parenti et al. [42].

#### 2.2.6. Determination of Fluoxetine Concentration

The quantification of fluoxetine was performed by HPLC-FLD using a Shimadzu LC system (Shimadzu Corporation, Kyoto, Japan) equipped with a LC-20AD pump, a DGU-20A 5R degasser, a CTO-10AS VP column oven, an SIL-20A HT automatic injector, and a RF-20A-XS fluorescence detector. The chromatographic separation was achieved using a Luna C18 column (150 × 4.6 mm, 5 μm particle size) (Phenomenex, Torrance, CA, USA), using the method described by Silva et al. [43]. The identification of the analyte was based on its retention time compared to a standard solution, and the quantification was performed using the external calibration curve method. A linear relationship between peak area and fluoxetine concentration was established in the range of 1–2000 μg∙L^−1^. The limit of detection (LOD) and limit of quantification (LOQ) were determined on the basis of the signal-to-noise ratio using the analytical response of 3 and 10 times of the background noise, respectively. The determined LOD and LOQ for fluoxetine were 4.60 and 1.53 × 10 μg∙L^−1^, respectively.

#### 2.2.7. Statistical Analysis

Statistical analysis was performed with MedCalc^®^ statistical software (version 12.5.0.0) (Medcalc Software, Ostend, Belgium). The normal distribution of results was tested using the Kolmogorov–Smirnov test, and the outliers’ presence was tested using the generalized extreme Studentized deviate (GESD). Removal efficiencies and laccase activities were described as mean ± standard deviation. Comparison of results was performed with a *t*-test of independent samples (Welch test for unequal variances). The statistical significance was defined as *p*-value < 0.05.

## 3. Results and Discussion

### 3.1. Results of Alder Sawdust Characterization

FT-IR analysis was performed to identify the functional groups that may act as binding sites in the biosorption of fluoxetine onto alder sawdust. The FT-IR spectrum is shown in Figure 2.

Five main peaks or absorption bands can be identified, and the respective wavenumber assignments are shown in Appendix A. In the functional group region, between 4000 and 2500 cm^−1^, bands A and B can be identified. The broad and blunt shape absorption band at about 3408 ± 0 cm^−1^ (band A) is assigned to O-H stretching vibrations of water or hydroxyl radicals [44,45,46,47,48]. The absorption band at around 2925 ± 0 cm^−1^ (band B) is assigned to saturated C-H stretching vibrations, more specifically to CH_2_ asymmetric and symmetric stretching vibrations [44,45,46,47]. In the region of double bonds, between 2000 and 1500 cm^−1^, the sharp shape absorption band at about 1650 ± 1 cm^−1^ (band C) is assigned to the C=O stretching vibration [44,47,48,49]. The absorption band at about 1539 ± 3 cm^−1^ (band D) is assigned to the aromatic C=C deformations [48]. The absorption band at about 1401 ± 25 cm^−1^ (band E) is assigned to CH_2_ and CH_3_ asymmetric deformations [44,48]. The absorption band at about 1238 ± 1 cm^−1^ (band F) is assigned to C-O stretching vibrations [48]. In the fingerprint region, below 1300 cm^−1^, the absorption band at about 1079 ± 52 cm^−1^ (band G) is assigned to C-O stretching vibrations [44,48,49]. Finally, absorption bands at about 668 ± 1 cm^−1^ and 601 ± 1 cm^−1^ (bands H and I, respectively) can be assigned to phenil ring substitutions, certifying the presence of aromatic nuclei in sawdust [48]. Sawdust is mainly composed of cellulose, hemicellulose (xylan and mannosan), and lignin, a non-negligible portion of lipids and waxes, bearing functional groups such as alcohol, ketone, and carboxylic groups [50]. The functional groups that might play an important role in the biosorption mechanism and may be responsible for fluoxetine-binding are hydroxyl, carboxyl, and ketone groups.

The pH affects the biosorption process, since it determines the speciation of the chemical species. A convenient index of the tendency of a surface to become positively or negatively charged as a function of pH is the value of pH that is required for the net charge of the adsorbent to be zero, the so-called zero charge point (pH_PZC_). The pH_PZC_ value determined for alder sawdust was 4.90 ± 0.01 (see Appendix A). For pH values lower than pH_PZC_, the surface charge of alder sawdust biomass is positive, and for pH values above pH_PZC_, the surface charge is negative [51].

Fluoxetine hydrochloride has an ionizable amino group with an acid dissociation constant (*p*Ka) of 9.8 (see Appendix A shows the species distribution diagram of fluoxetine hydrochloride as a function of pH. It is observed that there are three ranges, with boundaries defined by pH=pKa − 2 and pH=pKa+2 [52]. For pH values below 7.8, fluoxetine molecules are predominantly positively charged, whereas for pH values above 11.8, fluoxetine molecules are predominantly neutral [36,37]. In the pH range between 7.8 and 11.8, the solution is characterized by the coexistence of neutral and positive species. Thus, attraction forces can be predicted to occur mainly in the pH range of 4.9–9.8.

### 3.2. Results of Batch Assays

Batch assays were performed to assess the potential of the CMS to remove fluoxetine from aqueous solutions under non-sterile conditions and relate the efficiency to the laccase activity of the CMS. The blank experiments, run in parallel with batch assays, allowed us to verify that about 9% of fluoxetine concentration was abiotically removed, suggesting potential interactions of the PhC with the medium. All the removal efficiencies were calculated considering the blank as a reference.

The evolution of fluoxetine concentration and mean laccase activity of CMS tested in the 1st day and after fifteen days of storage (15th day) are shown in Figure 3, where $C_{0}$ (μg∙L^−1^) is the initial concentration and $C_{t}$ (μg∙L^−1^) is the concentration at a given time.

At the beginning of assays, the concentration of fluoxetine was 600 µg∙L^−1^, decreasing for both assays on the first 10 min of the reaction time. After that, the concentration of fluoxetine remained constant. The removal efficiencies achieved for CMS (1st day) and CMS (15th day) assays were 100.0 ± 0.0% and 84.6 ± 0.1%, respectively. The removal efficiencies are statistically different (*p*-value: 0.01), which may be related to the loss of laccase activity over fifteen days of storage, in the range 73.0 ± 18.7 U∙L^−1^ −19.2 ± 2.3 U∙L^−1^ (Figure 3b).

Hultberg et al. [31] studied the removal of diclofenac (2 µg∙L^−1^) from water using the same CMS (200 g∙L^−1^) and related it to the laccase activity of the CMS. A high diclofenac removal efficiency (80–90%) in a short period (5 min) of exposure to the CMS was observed. Similarly to this study, it was observed that diclofenac removal efficiency increased with laccase activity increase. The authors also studied the evolution of laccase activity over time. The results showed that laccase activity significantly decreased during cold storage (10 °C) within four days, reaching the highest level after incubation at 24 °C, immediately before initiation of fruiting body formation. This finding, also confirmed by other research [53,54], suggests a role of the physiological state of *Pleurotus ostreatus* in the exudation of laccases and that these enzymes have an important role in mycelial growth and nutrient acquisition before fructification. This implies, however, that the use of the CMS should be done immediately (in the very next day) at the WWTP to make this treatment option efficient.

Biosorption and CYP450 inhibition assays were performed to provide insights on the mechanisms involved in the fluoxetine removal by the CMS. The evolution of fluoxetine concentration and mean laccase activity of both assays are shown in Figure 4. The results of the CMS (15th day) assay are also shown in Figure 4 to establish a comparison.

As can be seen in Figure 4a, the fluoxetine removal profiles are similar, with a deep concentration decrease in the first 10 min. The removal efficiencies achieved for biosorption and CYP450 inhibition assays were, respectively, 44.7 ± 3.5% and 83.1 ± 0.7%. Nevertheless, the removal efficiencies of the CMS (15th day) and CYP450 inhibition assays were not statistically different (*p*-value: 0.14), suggesting that CYP450 enzymes’ contribution to fluoxetine removal was not relevant, which may be attributed to the fact that the CMS was used immediately before initiation of fruiting body formation. Although the assays were performed using the CMS with the same storage period, the laccase activities (Figure 4b) are statistically different (19.2 ± 2.3 and 8.9 ± 5.1 for CMS (15th day) and CYP450 inhibition assays, respectively) (*p*-value < 0.05), probably due to the CMS heterogeneity. On the other hand, the removal efficiencies of the CMS (15th day) and biosorption assays are statistically different (*p*-value: 0.03).

These results suggest that fluoxetine removal efficiency was highly impacted by biosorption (44.7 ± 3.5%) onto the CMS in synergy with LMEs activity. The biosorption of fluoxetine onto the CMS may also have provided a suitable environment for LMEs, probably favouring both biosorbed and dissolved fluoxetine degradation. In addition, some products that result from the alder sawdust and wheat bran degradation may have acted as laccase mediators, enhancing fluoxetine removal efficiency by the CMS. The biosorption of fluoxetine onto the sterilized CMS probably occurred due to its high organic content and functional groups available to interact with fluoxetine [55,56,57], as a consequence of its components, alder sawdust and wheat bran. As exposed in FT-IR analysis, alder sawdust possesses functional groups that may have a significant potential for fluoxetine binding. The biosorption mechanisms may also have been favoured by the fluoxetine’s hydrophobic character (log K_ow_ 4.17) and because fluoxetine molecules are predominantly positively charged at pH 6.46–6.94 (*p*Ka 9.8, see Appendix A), being within the pH range where attraction forces were predicted to occur (see Section 3.1).

For instance, Zhou et al. [57] demonstrated the ability of an SMS to remove trace concentrations of sulfamethyldiazine, sulfamethazine, sulfathiazole, and sulfamethazole from aqueous solutions.

It is important to note that, in this work, the substrate was previously pasteurized to decrease its microbiota and subsequently inoculate it with spawn. Therefore, the role of the substrate concerning *Pleurotus ostreatus* was mainly a support and nutrient source. However, it is an agricultural waste and, as such, usually has valuable microbiota (pasteurisation allows survival of spore-producers like *Bacillus* spp. that may also contribute favourably to the PhCs’ biodegradation [58]).

The evolution of residual fluoxetine concentration over the reaction time and mean laccase activity of CEE assays are shown in Figure 5. The results of the CMS (1st day) are also shown in the same figure to establish a comparison. Considering that laccase activities were not statistically different—73.0 ± 18.7 and 80.1 ± 3.5 U∙L^−1^ for CMS (1st day) and CEE assays, respectively (*p*-value: 0.40) (Figure 5b)—the higher removal efficiency achieved for CMS (1st day) assays (*p*-value < 0.05) can be attributed to the biosorption contribution, as discussed before. The fluoxetine removal profiles are also similar (Figure 5a). The removal efficiency achieved for CEE assays was 19.6 ± 0.9%.

The removal of non-phenolic compounds, such as fluoxetine, by extracellular LMEs is generally poor/unstable. The physicochemical properties of non-phenolic compounds appear to be a key reason for low degradation. Tadkaew et al. [59] inferred that the presence of electron-withdrawing functional groups (EWGs), such as amide, carboxylic, halogen, and nitro, in the molecular structure of compounds, seems to generate an electron deficiency, rendering them less susceptible to oxidative catabolism, thus exhibiting poor removal efficiency. On the opposite side, electron-donating functional groups (EDGs), such as amine, hydroxyl, alkoxy, alkyl, and acyl, render the compounds more prone to the electrophilic attack of aerobic bacteria, thus exhibiting high removal efficiency. Even though fluoxetine contains methyl, ether oxygen, and secondary amine EDGs, it is probable that the presence of a trifluoromethyl EWG may hinder laccase-catalyzed degradation (see molecular structure in Figure 1).

The laccase activity remained unchanged over the reaction time, which demonstrates that fluoxetine did not harm the stability of the laccase at the assay concentration. Laccases are glycoproteins, glycosylated blue oxidases containing four copper atoms in the active site classified into three types: T1, T2, and T3. The reactions catalyzed by these enzymes proceed by the mono-electronic oxidation of suitable reducing substrates to form reactive radicals at the expense of molecular oxygen, which is eventually reduced to form water molecules. The redox process takes place with the assistance of a cluster of the four copper atoms that form the catalytic core of the enzyme [60]. Due to the low redox potential of the T1 copper, laccase usually oxidizes phenols or phenolic lignin units due to matching redox features [61]. Non-phenolic substrates with redox potential above 1.3 V are more resistant to mono-electronic oxidation and are not oxidized by laccase directly [62]. The kind of substituents also has an influence on the oxidability. The EWGs, such as trifluoromethyl, may have decreased the possibility for oxidation [63,64,65]. This seems to limit the relevance of laccase in PhCs removal. However, the presence of redox-mediators in the reaction medium enables laccase to oxidize indirectly non-phenolic substrates [60].

Despite the low fluoxetine removal achieved in this study, when CEE was used, its application on PhCs’ removal from aqueous solutions has several advantages when compared to the use of live cultures (whole-cell cultures). The degradation of PhCs by WRF is a co-metabolic process, which occurs in the presence of an easily degradable substrate, typically glucose [25,66,67,68]. Extracted LMEs do not need the continuous addition of nutrients or to compete with other microorganisms such as bacteria. Moreover, the use of CEE allows the achievement of high reaction kinetics under mild temperature and pH conditions [68,69,70,71]. Compared to the use of available purified LMEs, the CEE have shown better PhCs removal efficiency, which is attributed to the natural mediators’ presence in the CEE [72,73,74]. In addition, the utilization of crude LMEs PhCs may considerably reduce the cost of the treatment process (as it avoids purification steps). The CEE may, however, contain significant nutrient levels that can increase the organic load of wastewater. The enzymatic degradation combined with other eco-friendly and cost-effective tertiary treatments, such as microalgae-based treatment, may be a possibility to solve this drawback [4,75,76].

Besides the presence of EWG and EDG groups in the molecular structure of fluoxetine and its high ORP, the performance of extracellular LMEs may have been affected by physicochemical properties of the reaction medium, namely by the temperature and pH. The temperature and pH values recorded in the batch assays (see Appendix A) only showed slight variations. Although the assays were performed considering the conditions expected for domestic wastewater after the secondary treatment, these conditions are outside the laccase optimal ranges.

The temperature acts on enzyme velocity. The effect produced is complex due to the protein nature of enzymes. The temperature increase above a specific value affects the tertiary structure of enzymes and the stability of the enzyme-substrate complex [77]. *Pleurotus ostreatus* secretes multiple isoforms of laccases. The different *Pleurotus ostreatus* isoforms have different optimal operating temperatures, 45–65 °C for POXA1w, 25–35 °C for POXA2, 35 °C for POXA3a and POXA3b, and 50–60 °C for POXC [21,69,78,79,80,81,82,83,84].

The pH of the reaction medium determines the charges of the enzyme amino acids. The charge’s change is related to conformational changes in the biocatalyst molecule. When the substrate has electrical charges, its approximation to the enzymes’ active center also depends on the residual charge involved in the binding. There is, therefore, an optimum pH value at which the formation of the enzyme-substrate complex is favoured, and the biological activity value is at a maximum [77]. Laccases are generally more stable at acidic pH. *Pleurotus ostreatus* isoforms have an acid optimal pH range for 2.6-dimethoxyphenol (2.6-DMP) oxidation, 3.0–5.0 for POXA1w and POXC, 6.5 for POXA2 [84], 5.5–6.5 for POXA3a and POXA3b [83], and 4.5 for POXC1b [21].

### 3.3. Results of Column Assays

The progression curves of residual fluoxetine concentration, laccase activity, and fluoxetine removal efficiency in the column outlet solution over time are shown in Figure 6, where $C_{0}$ (μg∙L^−1^) is the inlet solution and $C_{t}$ (μg∙L^−1^) is the concentration at a given time.

The temperature range of inlet and outlet solutions were, respectively, 13.0–25.0 °C and 25.5 ± 0.7 °C. The pH of inlet and outlet solutions remained stable at, respectively, 7.06 ± 0.01 and 6.99 ± 0.02.

As shown in Figure 6, fluoxetine was first detected in the outlet solution after 30 min of operation. The concentration recorded was 37 ± 2 µg∙L^−1^, corresponding to 5% of the fluoxetine concentration fed to the column. Thereafter, the residual fluoxetine concentration progressively increased until the end of the assay, evidencing a broad mass transfer zone. The theoretical exhaustion point (≈95% of the fluoxetine concentration fed to the column), 682 ± 25 µg∙L^−1^, was not achieved. During the eight hours of operation, the removal efficiency achieved was higher than 70%.

Batch assays showed that fluoxetine removal was due to the synergistic effect of LMEs activity, namely the laccase activity and biosorption onto the CMS. The removal efficiency decrease over the operation time may be related to the CMS saturation, the natural LMEs activity decay due to operation conditions, and the permanent washout of natural redox-mediators and LMEs by the flux passing through the column. The laccase activity recorded in the outlet solution was at a maximum at the beginning of the assay, 35.8 U∙L^−1^ at 10 min, continuously decreasing until stabilization after 150 min.

Fixed-bed reactors are the most used in continuous processes. They are kinetically more favourable than continuously stirred reactors and do not have high shear stresses due to mechanical stirring. However, there is the possibility of washout of both redox-mediators and LMEs, which may be overcome through enzyme immobilization or by integrating a membrane with a pore size smaller than the size of redox-mediators and LMEs molecules with the fixed-bed reactor. The commonly used approaches for enzyme immobilization (adsorption, covalent binding, entrapment/encapsulation, and cross-linking) imply changes in the enzymes structure or their microenvironment, which is reflected in changes in their catalytic activity and kinetic parameters. To guarantee the use of LMEs in their native form, a membrane may act as a barrier against redox-mediators and LMEs release, while transformation products can cross the membrane along with the treated solution [85]. Nevertheless, further studies are needed to assess the viability of the large-scale treatment process and its potential challenges, such as membrane fouling [86,87].

Furthermore, other factors may affect the LMEs activity and consequently the PhCs removal. Indeed, wastewaters are complex aqueous matrices that contain several organic and inorganic compounds described as laccase activity inhibitors: anions that interact with copper sites (e.g., azide, cyanide, thiocyanide, and fluoride), complexing agents that remove copper from the active site, metal ions (e.g., Hg^2+^), fatty acids, sulfhydryl reagents, hydroxyglycine, and quaternary ammonium cationic washing powders [88,89,90,91,92]. It is also important to note that, in this work, the research was focused on fluoxetine removal rather than degradation pathways and transformation products’ toxicity. The toxicity of the treated solution should be assessed before the decision for system scale-up in order to guarantee that the transformation products are not toxic. In previous studies, carbamazepine degradation by-products were found to be more toxic than their parent compound [12,25].

## 4. Conclusions

The ability of a lignocellulosic substrate colonized by the WRF *Pleurotus ostreatus* and its CEE to remove fluoxetine from aqueous solutions through batch and columns assays were investigated. The removal of fluoxetine by the CMS and CEE under non-sterile conditions was studied in batch systems. The contributions of biosorption and intracellular enzymes (i.e., CYP450) to the fluoxetine removal were also assessed. Fluoxetine removal using crude enzyme extracts (CEE) (extracellular content without CMS) was also performed. All assays were performed at room temperature and pH 7, considering the expected conditions for treated domestic wastewater (after secondary treatment). High removal efficiencies were achieved with the CMS (1st day) and CMS (15th day), 100.0 ± 0.0% and 84.6 ± 0.1%, respectively, in a short period (10 min). Assays with sterilized CMS and the addition of 1-ABT suggested that the high removal efficiencies achieved were due to the synergistic effects of extracellular LMEs activity, namely as laccase, coupled with sorption onto the CMS. The decrease of removal efficiency observed after fifteen days of storage was related to the loss of laccase activity, in the range 73.0 ± 18.7 U∙L^−1^ −19.2 ± 2.3 U∙L^−1^. A poor removal efficiency was achieved with CEE, 19.6 ± 0.9%, with a laccase activity of 80.1 ± 3.5 U∙L^−1^ (statistically equal to the CMS (1st day) assays), which was attributed to the presence of EWGs in the fluoxetine structure, to the high ORP of fluoxetine, and to the operation conditions (temperature and pH) outside the optimal range.

A fixed-bed column assay was performed using the CMS, which showed a good performance over the eight hours of continuous operation. The removal efficiency achieved was complete during approximately thirty minutes, then it decreased progressively over time to a final value of 70%, after 8 h of operation. However, the laccase activity was recorded in the treated solution, suggesting the need to couple, for instance, a membrane with a suitable pore size to avoid LMEs washout.

The results obtained suggest that CMS use is a good candidate for fluoxetine removal as a tertiary treatment of wastewaters, being an eco-friendly alternative that allows the valorization of waste from the food industry.

Nevertheless, further investigation is needed to better evaluate technical, economic, and environmental aspects before application of the CMS to real wastewater.

## Figures and Tables

**Figure 1 ijerph-19-02672-f001:**
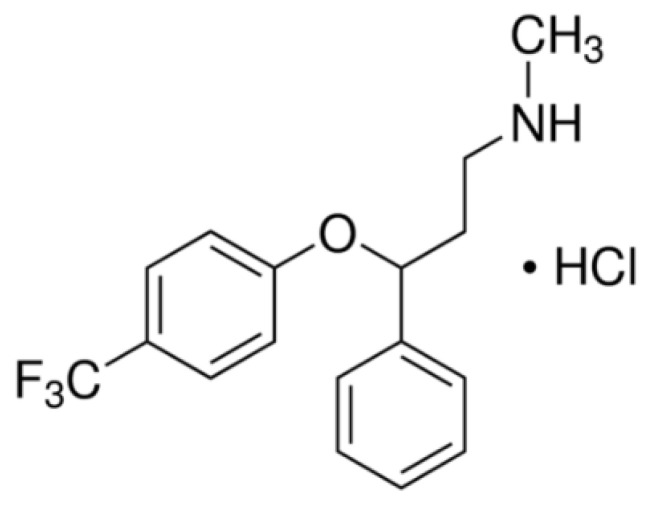
Molecular structure of fluoxetine hydrochloride.

**Figure 2 ijerph-19-02672-f002:**
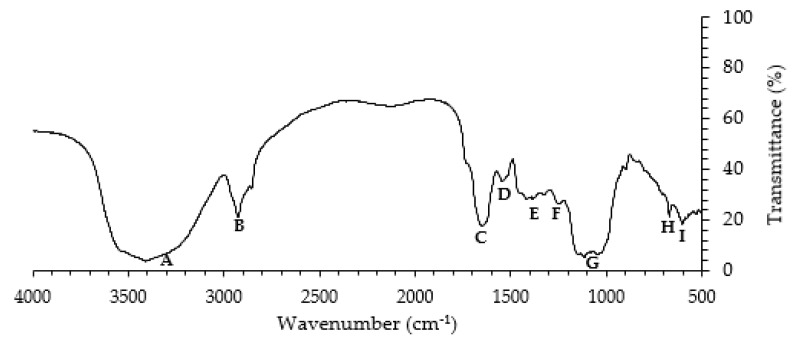
Fourier transform infrared (FT-IR) spectrum of alder sawdust.

**Figure 3 ijerph-19-02672-f003:**
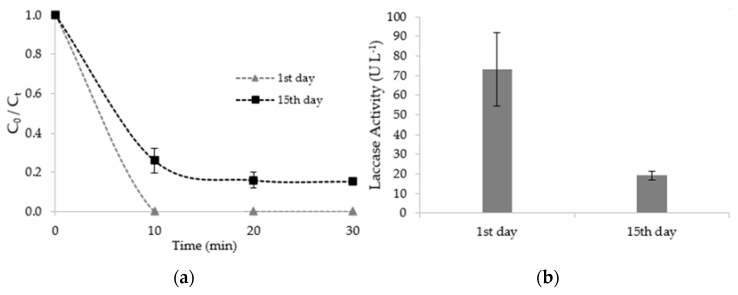
(**a**) Progression of fluoxetine concentration and (**b**) mean laccase activity of colonized mushroom substrate (CMS) at the first day and after fifteen days of storage.

**Figure 4 ijerph-19-02672-f004:**
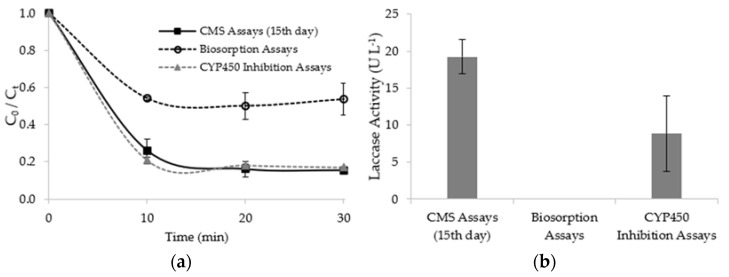
(**a**) Progression curves of residual fluoxetine concentration over time and (**b**) mean laccase activity of the CMS (after 15 days of storage), biosorption, and cytochrome P450 (CYP450) inhibition assays.

**Figure 5 ijerph-19-02672-f005:**
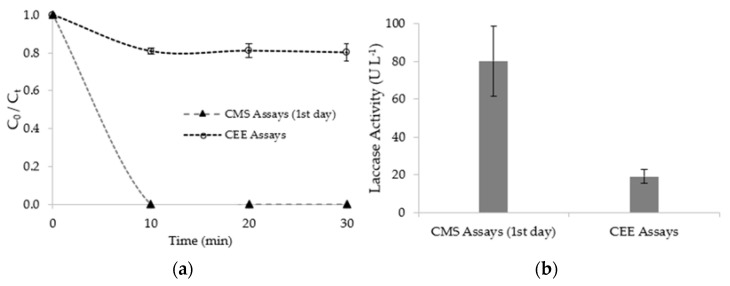
(**a**) Progression curves of residual fluoxetine concentration over time and (**b**) mean laccase activity of the CMS (at the first day) and crude enzyme extracts (CEE) assays.

**Figure 6 ijerph-19-02672-f006:**
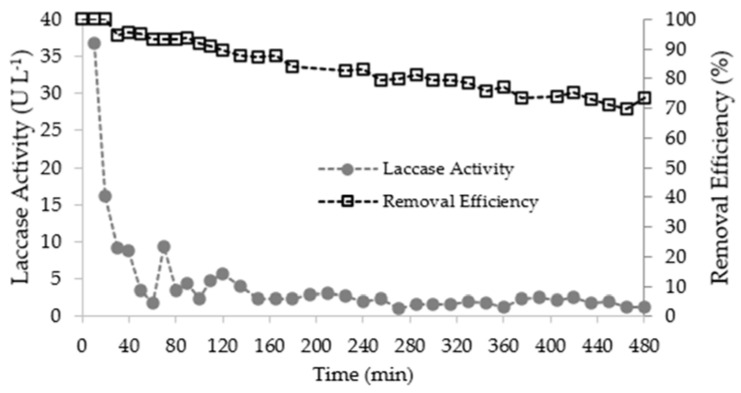
Progression curves of laccase activity and fluoxetine removal efficiency in the column outlet solution over time.

## Data Availability

Not applicable.

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
