# Peer review of "Fluoxetine Removal from Aqueous Solutions Using a Lignocellulosic Substrate Colonized by the White-Rot Fungus Pleurotus ostreatus"

_ijerph, 2022, doi:10.3390/ijerph19052672_

Round 1

Reviewer 1 Report

  1. The background should be mentioned in the Abstract.
  2. The English should be improved.
  3. The writing of section 2.1.2 should be revised.
  4. The section of 2.2 should be re-organized, some information could be placed in a supporting information.
  5. Why the peaks of A and B were described by “xx±0”? what is the meaning of the “±0”?
  6. In figure 2, there are some other peaks are not determined, why? Like the peaks at about 1550 cm-1, 1300 cm-1, 700 cm-1 and 500 cm-1. And the peaks for D ad E are not obvious.
  7. In Figure 4b, the error bar for the first day is so big, which is an error rather than an error bar. The plus and minus error is accounted about 60%.
  8. In Figure 5b, the plus and minus error is accounted above 100% for CYP 450. It is a error.
  9. The references should be reduced for a research article.

Author Response

The authors sincerely acknowledge the suggestions for improvement. Please find the responses in the attached file.

Reviewer 2 Report

The presented manuscript on the removal of fluoxetine with CMS reports interesting findings on the degradation/sorption capability of the tested substrate with regards to the investigated pollutant. The language style is excellent. The abstract precisely and adequately represents the study. The conclusions drawn from the results are appropriate. The research conducted seems original and the quality of the research seems high.

The main concern, however, is the significance of the content and, consequently, the length of the manuscript. Reference 38 by Hultberg et al. – one of the co-authors – seems to apply a very similar methodology (CMS, laccase activity correlation) to a different set of pollutants. The inhibition assays and the column tests are acknowledged as novel content; nonetheless, the mentioned concern prevails. Large fractions of the manuscript as well as tables and figures should be much more concise (for a detailed elaboration, see below).

Therefore, the authors should be encouraged to significantly shorten the manuscript and resubmit it as short communication, rather than full research paper.

Details:

  • The introduction until line 100 mostly repeats knowledge readily available in textbooks, be it the issue of organic micropollutants or the background on mycoremediation/WRF. A reduction is recommended.
  • The list of references is also excessive (e.g. [1-6], [8-13], but also other exuberant elaborations in the methods section and the results section).
  • Chapter 2 could generally be reduced, as most methods applied were carried out according to common practice.
  • Tab 1 could be reduced to text, leaving out the molecular structure.
  • Fig 1 is redundant, as the pKa is already mentioned in Tab 1.
  • The fourth level of headings (e.g. 2.2.1.1) should be removed, which allows for some reduction in length.
  • In line 193, the temperature range (20-25 °C) should be mentioned, which, along with the information on the pH in line 196 makes Tab 3 redundant.
  • 2.4: The moisture content is a standard method, on which the authors do not need to elaborate.
  • 2.5: The first sentence is sufficient.
  • 2.6 can be shortened. It is not necessary to report the calibration equation as well as Eq. 4.
  • Tab 2 is redundant, as the contained information is also stated in the text. The references are textbook knowledge.
  • Fig 3 is not necessary as the determination of the pHPZC is a standard method.
  • Large fractions of 3.2 can be shortened without jeopardizing the quality of the conducted research. References to citation 38 allows for further reduction.
  • 3 also shows potential for some shortening, especially after line 547.
  • Fig 4, 5, and 6 can be combined, as the also contain redundant information.
  • Fig 7 a is redundant, as the same information is plotted in Fig 7b.

Some further, rather minor aspects that could be improved:

  • 22: “The column assay…” should be replaced with “A column assay...”
  • Activated carbon should be mentioned as treatment alternative (line 56).
  • 61-64: The downsides of fungi-based technologies should be reported at some point.

Author Response

(The authors gave the same response as above.)

Round 2

Reviewer 1 Report

The paper had been improved.

Reviewer 2 Report

All suggestions have been implemented adequately and the quality of the manuscript has been improved significantly.